# Characteristics and predictors of Long COVID among diagnosed cases of COVID-19

**M. C. Arjun**[1], **Arvind Kumar Singh**[1]*, **Debkumar Pal**[1], **Kajal Das**[1], **Alekhya G.**[1], **Mahalingam Venkateshan**[2], **Baijayantimala Mishra**[3], **Binod Kumar Patro**[1], **Prasanta Raghab Mohapatra**[4], **Sonu Hangma Subba**[1]

**1** Department of Community Medicine and Family Medicine, All India Institute of Medical Sciences, Bhubaneswar, Odisha, India, **2** College of Nursing, All India Institute of Medical Sciences, Bhubaneswar, Odisha, India, **3** Department of Microbiology, All India Institute of Medical Sciences, Bhubaneswar, Odisha, India, **4** Department of Pulmonary Medicine and Critical Care, All India Institute of Medical Sciences, Bhubaneswar, Odisha, India

* arvind28aug@gmail.com

## Abstract

### Background

Long COVID or long-term symptoms after COVID-19 has the ability to affect health and quality of life. Knowledge about the burden and predictors could aid in their prevention and management. Most of the studies are from high-income countries and focus on severe acute COVID-19 cases. We did this study to estimate the incidence and identify the characteristics and predictors of Long COVID among our patients.

### Methodology

We recruited adult ($\geq$18 years) patients who were diagnosed as Reverse Transcription Polymerase Chain Reaction (RTPCR) confirmed SARS-COV-2 infection and were either hospitalized or tested on outpatient basis. Eligible participants were followed up telephonically after four weeks and six months of diagnosis of SARS-COV-2 infection to collect data on sociodemographic, clinical history, vaccination history, Cycle threshold (Ct) values during diagnosis and other variables. Characteristics of Long COVID were elicited, and multivariable logistic regression was done to find the predictors of Long COVID.

### Results

We have analyzed 487 and 371 individual data with a median follow-up of 44 days (Inter quartile range (IQR): 39,47) and 223 days (IQR:195,251), respectively. Overall, Long COVID was reported by 29.2% (95% Confidence interval (CI): 25.3%,33.4%) and 9.4% (95% CI: 6.7%,12.9%) of participants at four weeks and six months of follow-up, respectively. Incidence of Long COVID among patients with mild/moderate disease (n = 415) was 23.4% (95% CI: 19.5%,27.7%) as compared to 62.5% (95% CI: 50.7%,73%) in severe/critical cases(n = 72) at four weeks of follow-up. At six months, the incidence among mild/moderate (n = 319) was 7.2% (95% CI:4.6%,10.6%) as compared to 23.1% (95% CI:12.5%,36.8%) in severe/critical (n = 52). The most common Long COVID symptom was

**Data Availability Statement:** The anonymized data set has been uploaded to the public repository (figshare). The dataset can be accessed using the DOI: 10.6084/m9.figshare.21665618

**Funding:** The authors received no specific funding for this work.

**Competing interests:** The authors have declared that no competing interests exist.

fatigue. Statistically significant predictors of Long COVID at four weeks of follow-up were—Pre-existing medical conditions (Adjusted Odds ratio (aOR) = 2.00, 95% CI: 1.16,3.44), having a higher number of symptoms during acute phase of COVID-19 disease (aOR = 11.24, 95% CI: 4.00,31.51), two doses of COVID-19 vaccination (aOR = 2.32, 95% CI: 1.17,4.58), the severity of illness (aOR = 5.71, 95% CI: 3.00,10.89) and being admitted to hospital (Odds ratio (OR) = 3.89, 95% CI: 2.49,6.08).

## Conclusion

A considerable proportion of COVID-19 cases reported Long COVID symptoms. More research is needed in Long COVID to objectively assess the symptoms and find the biological and radiological markers.

## Introduction

COVID-19 was declared a pandemic in March 2020 [1]. Globally, 625 million people have been diagnosed, and around 6 million are reported dead due to COVID-19 [2]. Health systems worldwide are striving to stop the spread of the SAR-COV-2 virus and prevent death and complication due to COVID-19. Apart from acute illness, dialogues on the chronic effect of COVID-19 gained momentum as medical practitioners worldwide started reporting on post COVID complications even in mild cases [3]. It was observed that symptoms of COVID-19 either persist or new symptoms arise after a patient has recovered. Multiple nomenclatures began to appear to describe this condition which includes Long COVID, chronic COVID syndrome, long hauler COVID, post-acute sequelae of COVID-19, post-acute COVID19 syndrome etc [4,5].

Long COVID was discussed widely, and research was initiated to understand this phenomenon [6]. But there was no widely accepted definition for Long COVID, making it difficult to diagnose and treat the condition [7]. The National Institute for Health and Care Excellence (NICE) was among the first to come out with a rapid guideline to define Long COVID [8]. NICE defines Long COVID as signs and symptoms that continue or develop after acute COVID-19, including both ongoing symptomatic COVID-19 (from 4 to 12 weeks) and post-COVID-19 syndrome (12 weeks or more) [9]. Similarly, the Centers for Disease Control and Prevention (CDC) define Long COVID as a Post-COVID condition with a wide range of new, returning, or ongoing health problems people can experience four or more weeks after first being infected with the virus that causes COVID-19 [10]. WHO recently published a document defining the Long COVID based on the Delphi method [11].

Studies have shown that Long COVID can affect almost all systems in the body [12]. The most described in the literature are respiratory disorders, cardiovascular disorders, neurocognitive disorders, mental health disorders, metabolic disorders etc. Symptoms are in multitudes: including fatigue, breathlessness, cough, anxiety, depression, palpitation, chest main, myalgia, cognitive dysfunction ("brain fog"), loss of smell, etc. [12]. Newer symptoms are identified and included in the long COVID as evidence emerges [13]. Many initiatives have been launched to estimate the burden and characteristics of long COVID, especially in developed countries. The Office of National Statistics (ONS) in the United Kingdom gives an estimate of the prevalence and risk factors of long COVID using the national Coronavirus (COVID-19) Infection Survey (CIS) [14]. COVID Symptom study app is another data source [15]. National Institute of Health in the USA has also launched new initiatives to study Long COVID and is expected to

bring out more evidence [6]. Independent researchers are also working on understanding this phenomenon.

In India, significantly less attention has been given to the burden of Long COVID [16]. During this study's conception, there were no research papers available in peer-reviewed journals that measured the burden of Long COVID in India. As of today, India has more than 34 million cases of COVID-19 [17]. This can translate into a huge number of patients suffering from long COVID. Once the active cases come down, the already overstretched health systems can witness another public health crisis in the form of Long COVID. To mitigate this, we should have a clear idea about Long COVID to develop better management strategies. Post COVID care clinics are already functioning in some parts of the country [18]. The Government of India has also published a guideline for the management of post COVID sequelae [19]. But none of the systematic reviews on long COVID has included a study from India, and there is a wide evidence gap on this condition in India. Thus, it is pertinent that we undertake a study to measure the burden, the characteristics, and the predictors of long COVID in India to bring much-needed insight into this condition.

## Methodology

We estimated the incidence, characteristics, and predictors of Long COVID by following up a cohort of patients who were Revere Transcription polymerase chain reaction (RTPCR) positive COVID-19 cases. The study was conducted at All India Institute of Medical Sciences (AIIMS) Bhubaneswar, a tertiary care government hospital and research institute. The study population included adult cases (age ≥18 years) of COVID-19 who were diagnosed with RTPCR test from AIIMS Bhubaneswar from April to September 2021. We did not test for the variant of COVID-19 but based on the Indian SARS-CoV-2 Genomics Consortium (INSA-COG) data the predominant COVID-19 variant circulating in the community during the study period was Delta (B.1.617.2) [20]. Individuals less than 18 years and pregnant women were excluded.

We accessed the AIIMS Bhubaneswar COVID-19 screening OPD database and records of patients admitted due to COVID-19. The database was cleaned by removing individuals with missing phone numbers, patients who expired, and those less than 18 years. As per the operational definition based on NICE guidelines, these individuals were contacted through telephone after four weeks and six months from the date of their COVID-19 diagnosis. After taking verbal consent, a detailed telephonic interview was conducted to record the socio-demographic details, past medical history including chronic disease and substance use, acute manifestations of COVID-19, and the treatment received. Participants self-reported their height and weight, and Body Mass Index (BMI) was derived. BMI was classified based on World Health Organization criteria (W.H.O) [21]. Data on COVID-19 vaccination history was also collected. This was followed by self-reported Long COVID symptoms and their characteristics which included fatigue, cough, loss of taste and smell, cognitive dysfunction (Brain fog), etc., and an open-ended question. The interview questions were adapted from the W.H.O Global COVID-19 Clinical Platform Case Report Form (CRF) for Post COVID condition (Post COVID-19 CRF) [22]. All the data in this study was collected by the post-graduate student authors. Pre-testing of the questionnaire was done, and supervised calls were made before the beginning of actual data collection. The data collected during telephonic interviews were directly entered into EpiCollect5 app. An individual who could not be contacted after two attempts were excluded. The RTPCR cycle threshold (Ct) values during diagnosis of COVID-19 were retrieved from the hospital database to study its association with Long COVID symptoms.

## Sample size and statistical analysis

The primary objective of this study was to estimate the percentage of participants who reported Long COVID symptoms. Apart from the overall percentage, we planned to estimate the percentage in the subgroups of COVID-19 patients separately, classified based on the severity of the acute COVID-19 disease. For mild/moderate acute COVID-19 subgroup, we assumed the percentage of Long COVID to be 20% based on previous literature [14]. A relative precision of 20% was used to derive a sample size of 400 for mild/moderate acute COVID-19 subgroup. Using the same approach, we also calculated the required sample size in severe/critical acute COVID-19 subgroup by estimating the prevalence of Long COVID to be 50%. The estimated sample size came to be 100 in the severe/critical subgroup. Thus, an overall 500 patients were targeted to be enrolled in the study.

Data was collected using EpiCollect5 and imported into Microsoft Excel for cleaning. The data was analyzed in statistical software R (version 3.6.3) and STATA version 16 (StataCorp, College Station, Texas 77845 USA). Incidence of Long COVID was determined by the number of participants who self-reported any of the Long COVID symptoms. The self-reported characteristics of symptoms were also given as proportions. The data were analyzed separately for mild to moderate and severe to critical acute COVID-19 patients. Logistics regression was used to find the predictive factors of Long COVID at the four weeks follow-up. Demographic variables, medical history including variables related to acute COVID-19 disease and COVID-19 vaccination were included in the logistic regression model based on the previous literature. Statistical significance for univariable analysis was set at p-value less than 0.05. Multivariable logistic regression was done to obtain an adjusted odds ratio with a 95% confidence interval. Clinically significant variables were added to the multivariable model.

## Ethical issues

The institutional ethics committee (IEC) of AIIMS Bhubaneswar granted ethical approval before starting the study (IEC Number: T/IM-NF/CM&FM/21/37). The study was explained to each individual, and a telephonic verbal consent was taken before starting data collection. The consent process was approved by the Institutional Ethics Committee. After data collection, if the participant was found to have symptoms of Long COVID, they were referred to Long COVID OPD in the department of Pulmonary Medicine, AIIMS Bhubaneswar.

## Results

We listed 698 COVID-19 RTPCR positive cases from April to September 2021, out of which 189 patients could not be contacted or enrolled. A total of 509 individuals were eligible to be included in the study. Consent was denied by nine participants, and thus a total of 500 interviews were conducted successfully at four weeks of follow-up. On the preliminary evaluation of data, thirteen entries had wrong dates and were dropped from the final analysis. A final sample of 487 individuals was analyzed at a median follow-up of 44 days (IQR = 39,47). The 487 participants were further followed up after six months. A total of 371 participants was successfully interviewed with a median follow-up of 223 days (IQR: 195,251) (Fig 1).

The mean age of the study participants was 39 years (SD = 15 years), ranging from 18 to 88 years. One hundred ninety-nine (40.9%) participants were female, and the majority were college graduates. Most of the participants were either unemployed or students or homemakers with no earnings. Thirty participants (6.2%) reported being in a job involving COVID-19 management. The majority of participants had normal BMI. (Table 1) Eighteen participants (3.7%) reported that they had COVID-19 before the current episode. Around 10% had a history of pre-existing Diabetes or Hypertension. Few participants gave the history of other

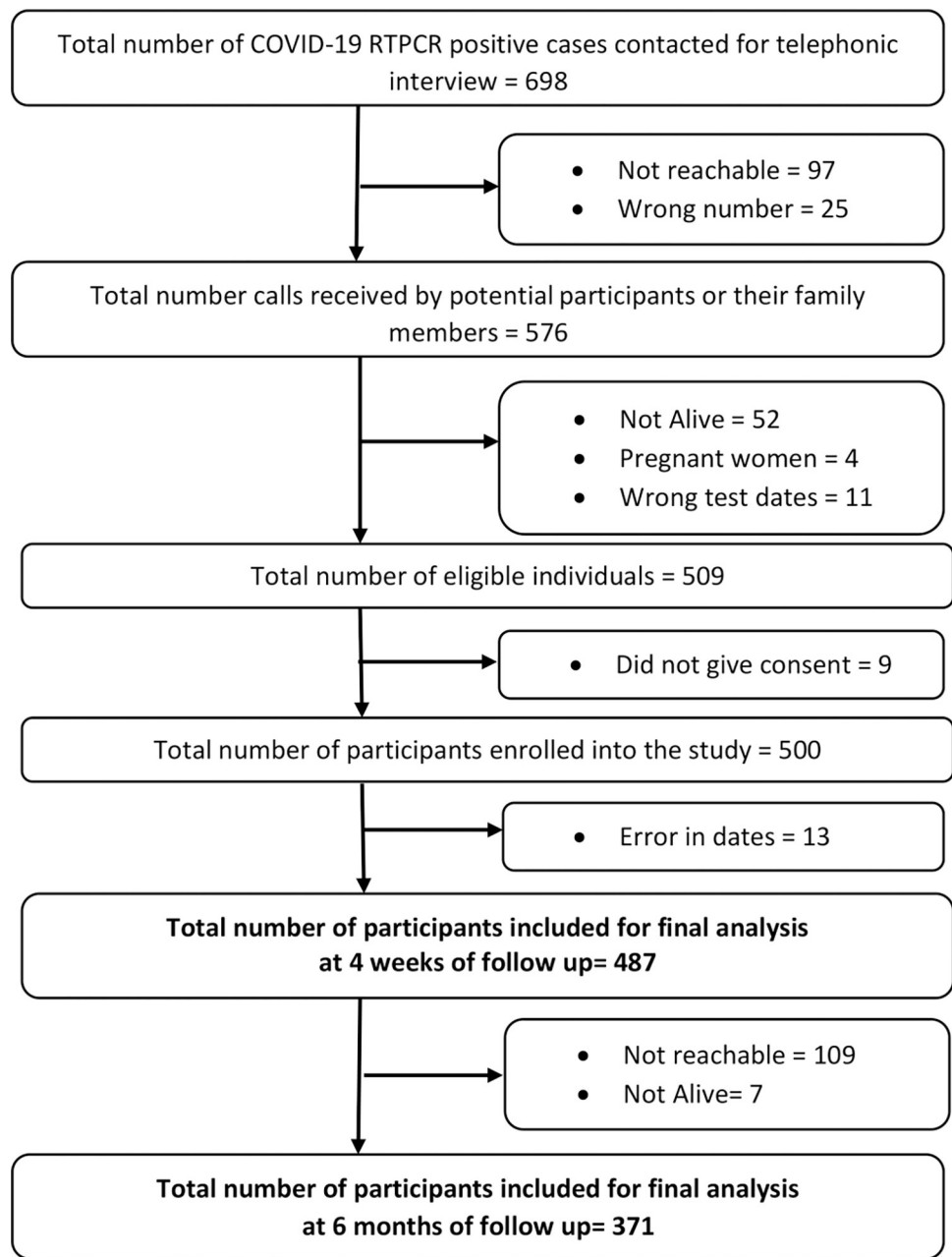

**Fig 1. Flow chart showing the selection of study participants and follow-up at four weeks and six months.**

comorbidities like asthma, tuberculosis, anxiety, cancer, or other chronic diseases, and none reported depression. A single question was used to record any type of self-reported substance use, and 54 (11.1%) participants gave the history of some form of substance use which included alcohol and tobacco. Two doses of vaccine were taken by 287 (58.9%) participants, one dose by 81 (16.6%), and there were 119 (24.5%) who had not been vaccinated at all. The majority of the sample had taken Covaxin. Very few participants reported having side effects post-vaccination.

Clinical features of the participants revealed that majority of them had 1 to 4 symptoms during the acute phase, and the most common symptoms were Fever and Cough. According to W.H.O Post COVID-19 Case Report Form criteria, 415 (85.2%) had mild to moderate and

**Table 1. Sociodemographic characteristics, past medical history, and vaccination status of participants (n = 487).**

| Socio-demographic characteristics | | n (%) |
|---|---|---|
| **Variable** | | **n (%)** |
| Age (Mean (SD); Range) | | 39 (15); 18 to 88 |
| Females | | 199 (40.9) |
| Education | Illiterate/No formal education | 24 (4.9) |
| | Studied up to 10 std or below | 116 (23.8) |
| | Higher secondary | 75 (15.4) |
| | College Graduate | 218 (44.8) |
| | Post-graduate and above | 54 (11.1) |
| Current Occupation | Unemployed/Student/Homemaker | 205 (42.1) |
| | Professionals/Technical/Administrators | 149 (30.6) |
| | Skilled and Unskilled Manual laborer | 54 (11.1) |
| | Retired | 11 (2.3) |
| | Other | 68 (13.9) |
| Occupation involving COVID-19 management | | 30 (6.2) |
| BMI (n = 484) | Underweight (<18.5) | 24 (5) |
| | Normal (18.5–24.9) | 284 (58.7) |
| | Overweight (25.0–29.9) | 153 (31.6) |
| | Obese (≥30.0) | 23 (4.7) |
| **Past Medical History** | | |
| History of COVID-19 before the current episode | | 18 (3.7) |
| Diagnosed to have Diabetes | | 58 (11.9) |
| Diagnosed to have Hypertension | | 51 (10.5) |
| Diagnosed to have Anxiety | | 3 (0.6) |
| Diagnosed to have Asthma | | 15 (3.1) |
| Diagnosed to have Tuberculosis | | 5 (1) |
| Diagnosed to have Cancer | | 19 (3.9) |
| Diagnosed to have other medical condition | | 53 (10.9) |
| Participants who gave history of substance use | | 54 (11.1) |
| Smoking status | Current smoker | 16 (3.3) |
| | Former (Not smoked more than one year) | 7 (1.4) |
| Chewable tobacco | | 32 (6.6) |
| Alcohol use | | 19 (3.9) |
| **COVID-19 Vaccination status** | | |
| Participants who received two doses of COVID-19 vaccine | | 287 (58.9) |
| Participants who received one dose of COVID-19 vaccine | | 81 (16.6) |
| Participants who did not receive COVID-19 vaccine | | 119 (24.5) |

68 (14%) had severe disease, and four participants (0.8%) had become critical. Most of the participants, 377 (77.4%), underwent home-isolation and were treated as Outpatient and 110 (22.6%) were hospitalized (Table 2).

At four weeks of follow-up, the overall incidence of Long COVID was 29.2% (95% CI: 25.3%,33.4%) with 142 individuals reporting it. Subgroup analysis revealed that incidence of Long COVID was 62.5% (95% CI: 50.7%,73%) among severe/critical cases (n = 72), which was significantly higher than among mild/moderate cases (n = 415) at 23.4% (95% CI: 19.5%,27.7%). (Fig 2) Among participants who were asymptomatic during the acute phase of COVID-19 (n = 111), only six reported Long COVID symptoms.

**Table 2. Clinical features and management of acute illness of COVID-19 among participants (n = 487).**

| Total number of symptoms reported by each participant | | n (%) |
|---|---|---|
| | No symptoms | 111 (22.8) |
| | 1 to 4 symptoms | 312 (64.1) |
| | 5 or more symptoms | 64 (13.1) |
| Most common symptoms reported | | |
| | Fever | 316 (64.9) |
| | Cough | 221 (45.4) |
| | Body ache | 89 (18.3) |
| | Breathing difficulty | 78 (16) |
| | Loss of smell | 63 (12.9) |
| | Loss of taste | 60 (12.3) |
| Severity of acute illness | Mild/Moderate–Did not receive oxygen | 415 (85.2) |
| | Severe- Required oxygen or was told you required oxygen | 68 (14) |
| | Critical–Received invasive ventilation t | 4 (0.8) |
| Care received during acute illness | Home isolation | 377 (77.4) |
| | Admitted to hospital | 110 (22.6) |

At four weeks of follow-up the most common symptom reported was Fatigue 92 (64.8%), followed by Cough 46 (32.4%). Only three participants reported cognitive dysfunction or Brain fog. Limitation of daily activity following Long COVID was not reported by the majority, but 41 (28.9%) participants reported having some activity limitation. Out of the 142 participants who self-reported Long COVID, 131 (92.3%) perceived the symptoms to be not severe, whereas 11 (7.7%) experienced the symptoms a lot. Health care practitioners were consulted for Long COVID by 49 (34.5%) participants (Table 3).

At six months, we followed up 371 out of 487 participants (Lost to follow-up 23.8%). The incidence of Long COVID reported was 9.4% (95% CI: 6.7%,12.9%) with a median follow-up of 223 days (IQR: 195,251). The incidence among mild/moderate (n = 319) was 7.2% (95% CI:4.6%,10.6%) as compared to 23.1% (95% CI:12.5%,36.8%) in severe/critical (n = 52). (Fig 2) Fatigue was the most common symptom. (Table 3) Between the first and second follow-up, 15 participants newly reported Long COVID symptoms, 151 participants received additional vaccination, and 5 participants were newly diagnosed with diabetes. During the follow-up period, seventeen participants were again diagnosed with COVID-19, which was mild/moderate in severity, out of which only one participant reported Long COVID.

We analyzed the predictors for Long COVID at 4 weeks of follow-up. Analysis revealed that age, sex, occupation, BMI, history of substance use, and Cycle threshold (Ct) values were not significantly associated with Long COVID. Pre-existing medical conditions (adjusted Odds ratio (aOR) = 2.00 (95% CI: 1.16,3.44)), receiving two doses of COVID-19 vaccination (aOR = 2.32 (95% CI: 1.17,4.58)), having more severe COVID-19 disease (aOR = 5.71 (95% CI: 3.00,10.89)) and having a greater number of symptoms during acute phase of COVID-19 disease were significantly associated with Long COVID. Admission to hospital during the acute phase of disease was significantly associated with Long COVID (Odds ratio = 3.89 (95% CI: 2.49,6.08)); however, this variable was not included in the multivariable model due to the colinear relation with the severity of the disease. (Table 4) The odds ratio of Long COVID for acute COVID-19 severity remained similar for subgroups based on COVID-19 vaccination status. For two doses of COVID-19 vaccination, the odds ratio of Long COVID for acute COVID-19 severity was 7.5 (95% CI: 3.1,18.4), and for zero to one dose of COVID-19

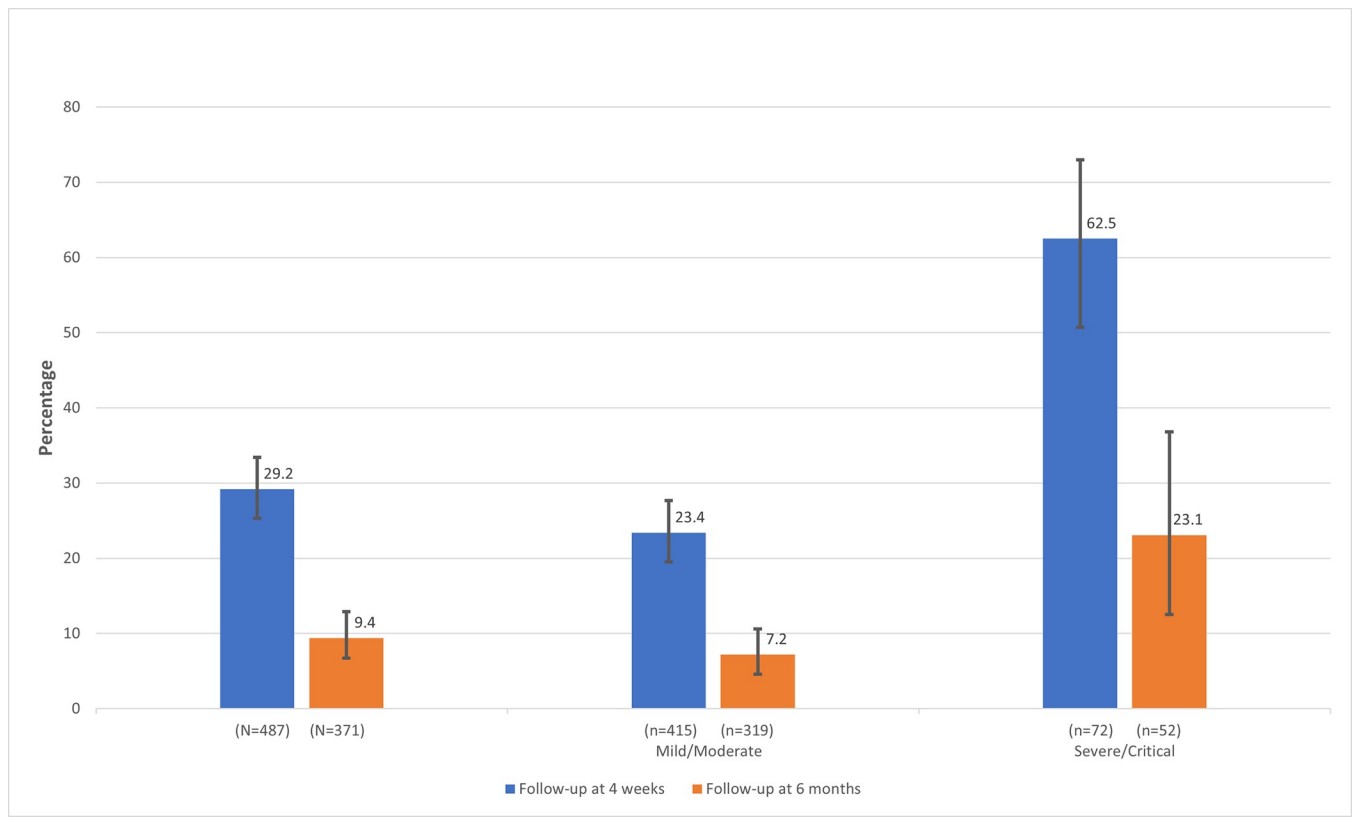

**Fig 2. Percentage of self-reported Long COVID symptoms at four weeks and six months of follow-up.**

**Table 3. Self-reported Long COVID symptoms and its features at four weeks and six months follow-up.**

| Variable | | At 4 weeks (N = 487) n (%) | At 6 months (N = 371) n (%) |
|---|---|---|---|
| Self-reported Long COVID symptoms | | **n = 142** (29.2) | **n = 35** (9.4) |
| Most common Self-reported Long COVID symptoms | Fatigue | 92 (64.8) | 19 (54.3) |
| | Cough | 46 (32.4) | 6 (17.1) |
| | Breathing difficulty | 24 (16.9) | 10 (28.6) |
| | Chest pain | 12 (8.4) | 2 (5.7) |
| | Loss of taste | 6 (4.2) | 0 |
| | Loss of smell | 4 (2.8) | 1 (2.9) |
| | Brain fog | 3 (2.1) | 0 |
| | Palpitation | 3 (2.1) | 1 (2.9) |
| | Anxiety | 3 (2.1) | 1 (2.9) |
| | Depression | 0 | 4 (11.4) |
| Limitation of activity | Activity limited a lot | 10 (7) | 0 |
| | Activity limited a little | 41 (28.9) | 16 (45.7) |
| | No activity limitation | 91 (64.1) | 19 (54.3) |
| Perceived severity of Long COVID symptoms | Not severe | 131 (92.3) | 33 (94.3) |
| | Severe | 11 (7.7) | 2 (5.7) |
| Consulted a health care practitioner | | 49 (34.5) | 8 (22.9) |

**Table 4. Predictors of self-reported Long COVID symptoms at four weeks follow-up.**

| Variable | | Univariable logistic regression | | Multivariable logistic regression | |
|---|---|---|---|---|---|
| | | Odds Ratio (95% CI) | p value | Adjusted Odds Ratio (95% CI) | p value |
| Age Categories | 18 to 45 years | Reference | - | Reference | - |
| | 46 to 59 years | 1.46 (0.91,2.36) | 0.12 | 1.24 (0.68,2.29) | 0.48 |
| | 60 years & above | 1.46 (0.79,2.67) | 0.22 | 1.08 (0.48,2.43) | 0.86 |
| Sex | Male | Reference | - | Reference | - |
| | Female | 1.33 (0.89,1.97) | 0.16 | 1.29 (0.74,2.25) | 0.36 |
| Occupation | Unemployed/Student/Homemaker | Reference | | Reference | |
| | Professional/Technical/ Administrative/ Managerial | 1.32 (0.84,2.08) | 0.23 | 1.79 (0.96,3.33) | 0.06 |
| | Skilled/Unskilled manual | 0.65 (0.31,1.34) | 0.24 | 0.82 (0.32,2.09) | 0.68 |
| | Other | 0.98 (0.55,1.74) | 0.94 | 1.15 (0.53,2.48) | 0.73 |
| BMI | Underweight (<18.5) | Reference | - | Reference | - |
| | Normal or lean (18.5–24.9) | 2.13 (0.71,6.44) | 0.18 | 1.58 (0.39,6.47) | 0.52 |
| | Overweight (25.0–29.9) | 2.35 (0.76,7.26) | 0.14 | 1.49 (0.35,6.26) | 0.58 |
| | Obese (≥30.0) | 1.05 (0.23,4.82) | 0.95 | 0.56 (0.09,3.44) | 0.53 |
| History of substance use | | 0.75 (0.39,1.44) | 0.38 | 0.95 (0.41,2.16) | 0.89 |
| Past history of COVID-19 | | 0.93 (0.33,2.66) | 0.90 | 0.66 (0.20,2.15) | 0.49 |
| Pre-existing medical condition | | 1.69 (1.12,2.55) | **0.01** | 2.00 (1.16,3.44) | **0.01** |
| COVID-19 vaccination | Not vaccinated | Reference | - | Reference | - |
| | Completed 1 dose | 1.30 (0.66,2.55) | 0.45 | 1.88 (0.84,4.22) | 0.13 |
| | Completed 2 doses | 2.05 (1.23,3.42) | **0.01** | 2.32 (1.17,4.58) | **0.01** |
| Number of COVID-19 symptoms | No symptoms | Reference | - | Reference | - |
| | 1 to 4 symptoms | 9.40 (3.99,22.09) | **<0.001** | 6.88 (2.74,17.23) | **<0.001** |
| | 5 or more symptoms | 12.77 (4.89,33.37) | **<0.001** | 11.24 (4.00,31.51) | **<0.001** |
| Severity of COVID-19 disease | Mild/Moderate | Reference | - | Reference | |
| | Severe/Critical | 5.46 (3.22,9.27) | **<0.001** | 5.71 (3.00,10.89) | **<0.001** |
| Care received during COVID-19 disease | Home Isolation | Reference | - | - | - |
| | Admitted to hospital | 3.89 (2.49,6.08) | **<0.001** | - | - |
| Cycle threshold | E Gene/N Gene (n = 442) | 0.98 (0.94,1.02) | 0.38 | - | - |
| | ORF1a/ORF1b/N/N2 Gene (n = 378) | 0.99 (0.95,1.03) | 0.53 | - | - |

vaccination, the odds ratio was 7.4 (95% CI: 3.5,15.8). This analysis rules out the possibility of interaction between COVID-19 vaccination and acute COVID-19 severity.

## Discussion

Long COVID is studied extensively all around the world, but research from India is limited. A recently published living systematic review has identified important research gaps, which includes paucity of evidence from low to middle-income countries and in people who were not hospitalized [23]. Both these research gaps are addressed in our study.

The overall incidence of Long COVID in our study was 29.2%, with a median follow-up period of 44 days. This is comparable to the Office of National Statistics (UK) estimates based on their National Coronavirus (COVID-19) Infection Survey. The survey estimates that around 1 in 5 respondents testing positive for COVID-19 exhibit symptoms for five weeks or longer, i.e., 21% (CI: 19.9,22.1) [14]. In mild to moderate cases, the symptoms of Long COVID were 23.4% in our study, after four weeks of COVID-19 infection. A study from India reported

Long COVID symptoms in mild COVID-19 to be 22.6% (prevalence of fatigue), albeit with a low sample size [24]. Similarly, another study from Northern India, which followed up the patients from a tertiary care hospital, estimates that 22% had Long COVID [25]. In severe to critical cases with a sample size of 72, our study estimated an incidence of 62.5%. These estimates are similar to a study from India published in pre-print server, which reported dyspnea in 74.3% and fatigue and disturbed sleep in more than 50% of patients after 30 to 40 days of recovery [26]. Another study in pre-print, which estimated Long COVID in hospitalized patients of North India, gave an estimate of 40.3% after 4 to 6 weeks follow-up [27]. High prevalence of Long COVID symptoms in severe and hospitalized cases are reported from multiple studies from all over the world [28,29].

We followed the same cohort of 487 individuals for a median of 223 days. There were 23.8% lost to follow-up. The Long COVID incidence reported in the six months follow-up was considerably low at 9.4% compared to the 29.2% reported at four weeks. At the six months follow-up, many participants did not report the symptoms of Long COVID that they reported during the four weeks follow-up. The incidence of 9.4% of Long COVID at six months is also very low when compared to other cohort studies from different parts of the world [30–33].

The most common Long COVID symptoms found in our study was fatigue. This is similar to other studies from India [24–26,34]. The Self-reported symptoms in the COVID Symptom Study app and the National Coronavirus (COVID-19) Infection Survey (CIS) from the United Kingdom has also recorded that fatigue is the most common symptom reported [14,15]. Multiple systematic reviews and meta-analysis on Long COVID have listed fatigue as the most common or among the first three Long COVID symptoms [12,23,35–37]. A recent study from India reported fatigue to be present even after three months of recovery from COVID-19 [38]. Although fatigue was self-reported in this study, a consistent finding in multiple studies indicates that fatigue is, in fact, the most common of Long COVID symptoms [39].

Predictors of Long COVID are important because it helps to prioritize the at-risk population and design interventions. In our study, one of the strongest predictors of Long COVID was the severity of COVID-19 disease and hospital admission. This is intuitive because the chances of having persistent symptoms after four weeks post-infection can be higher if the disease is severe. This is backed up by a systematic review which found that hospitalization during the acute infection (odds ratio [OR] 2·9, 95% CI 1·3–6·9) was the most significant predictor of developing the post-COVID syndrome [36]. Similar to the severity of the COVID-19 disease, having more than one symptom during the acute phase of COVID-19 disease was associated with Long COVID. This finding is similar to the COVID symptoms app study based on self-reported symptoms [15]. Another important predictor of Long COVID was the presence of pre-existing conditions like diabetes and hypertension. A study from India and a systematic review on this topic has found a similar and strong association between the pre-existing condition and Long COVID [26,35]. Age and sex, which was commonly found to be associated with Long COVID was not a significant predictor in our study. Cycle threshold (Ct) values of two genes were also not a significant predictor of Long COVID.

An observational paradox in our study was that the participants who took two doses of COVID-19 vaccination had higher odds of developing Long COVID. It could be due to better survival in vaccinated individuals who may continue to exhibit symptoms of COVID-19 disease. We could not find any interaction effect of COVID-19 vaccination and acute COVID-19 severity on causing Long COVID. This association might have also arisen due to Collider bias [40]. The Collider bias might have operated in this case since the sample included only COVID-19 positive tested patients who accessed the hospital (healthcare workers included) making the sample inherently biased to derive such conclusions. A rapid review by UK Health Security Agency has concluded that vaccinated people are less likely to report Long COVID

symptoms [41]. Although most studies show a negative association of COVID-19 vaccination and Long COVID, a recent study of 13 million people has reported that Long COVID risk falls only slightly after vaccination [42]. The negative association of COVID-19 vaccination and development of Long COVID is reiterated with the recent systematic review which concluded with low level of evidence that vaccination before SARS-CoV-2 infection could lower the risk for development of Long COVID [43].

The strength of this study is that all the cases of COVID-19 were diagnosed with RTPCR, and there is minimal risk of misclassification. The questionnaire used to capture the Long COVID was adapted from the standard case reporting format recommended by W.H.O. The data were collected by doctors involved in patient care and which improves the validity of the findings. Our study also had limitations. The Long COVID symptoms were all self-reported, and thus objective assessment of symptoms like fatigue was not done. Telephonic interviews precluded us from collecting additional information like clinical and radiological examination for correlating with the findings. Telephonic interview can also introduce recall bias especially with collecting information regarding confounding variables like severity of COVID-19 disease, vaccination status etc. The cause of death of fifty-two individuals who were not alive during the time of data collection was not enquired, and their death may have been related to Long covid complications. Also, we could not compare the baseline characteristics of participants with non-responders who were not included in the study, to assess for response bias.

In developed countries, many large-scale cohort studies are undertaken to understand this phenomenon [33,44]. Similar studies on Long COVID are lacking in India, and our research community should bridge this gap. We need more research into Long COVID to objectively assess the symptoms, to monitor the symptoms for a longer duration, and to study the biological and radiological markers, which can lead to better treatment guidelines and comprehensive management of COVID-19 disease.

## Supporting information

**S1 Checklist. STROBE statement—Checklist of items that should be included in reports of cross-sectional studies.**
(DOCX)

## Author Contributions

**Conceptualization:** M. C. Arjun, Arvind Kumar Singh.

**Data curation:** M. C. Arjun, Arvind Kumar Singh, Debkumar Pal, Kajal Das, Alekhya G.

**Formal analysis:** M. C. Arjun.

**Investigation:** M. C. Arjun, Arvind Kumar Singh, Debkumar Pal, Kajal Das, Alekhya G., Mahalingam Venkateshan.

**Methodology:** M. C. Arjun, Arvind Kumar Singh.

**Project administration:** M. C. Arjun, Arvind Kumar Singh, Debkumar Pal.

**Resources:** M. C. Arjun, Arvind Kumar Singh, Mahalingam Venkateshan, Baijayantimala Mishra, Binod Kumar Patro, Prasanta Raghab Mohapatra, Sonu Hangma Subba.

**Supervision:** M. C. Arjun, Arvind Kumar Singh, Baijayantimala Mishra, Binod Kumar Patro, Prasanta Raghab Mohapatra, Sonu Hangma Subba.

**Validation:** Arvind Kumar Singh, Debkumar Pal.

Writing – **original draft:** M. C. Arjun.

Writing – **review & editing:** M. C. Arjun, Arvind Kumar Singh, Debkumar Pal, Prasanta Raghab Mohapatra, Sonu Hangma Subba.

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
