## [Decision Letter · Decision Letter 0]

7 Oct 2022

PONE-D-22-01446Prevalence, characteristics, and predictors of Long COVID among diagnosed cases of COVID-19PLOS ONE

Dear Dr. Arvind Kumar Singh,

Thank you for submitting your manuscript to PLOS ONE. After careful consideration, we feel that it has merit but does not fully meet PLOS ONE’s publication criteria as it currently stands. Therefore, we invite you to submit a revised version of the manuscript that addresses the points raised during the review process. We appreciate your study which is an interesting study. However, the manuscript does not meet the publication criteria as the study design and statistical analysis should be appropriate and data should be described in sufficient detail. Please carefully consider and respond all of the reviewers’ comments, criticisms and suggestions. Please submit your revised manuscript by Oct 17 2022 11:59PM. If you will need more time than this to complete your revisions, please reply to this message or contact the journal office at plosone@plos.org. Please include the following items when submitting your revised manuscript:

We look forward to receiving your revised manuscript.

Kind regards,

Vipa Thanachartwet, M.D.

Academic Editor

PLOS ONE

Journal Requirements:

2. Please ensure that you include a title page within your main document. You should list all authors and all affiliations as per our author instructions and clearly indicate the corresponding author

Additional Editor Comments (if provided):

There are some important points raised as follows:

This study aims to estimate the prevalence and identify the characteristics and predictors of Long COVID among patients with acute COVID-19. Long COVID occur following acute COVID-19 and this condition develop during a particular time period, i.e., a new case, therefore, this study should determine the incidence rather than the prevalence. The median follow-up time of participants was 44 days in the results (page 7), however the follow-up time was too short for determining the occurrence of Long COVID.In methodology (page 5), “these individuals were contacted through telephone after four weeks from the date of their COVID-19 diagnosis. After taking verbal consent, a detailed telephonic interview was conducted to record the socio-demographic details, past medical history including chronic disease and substance use, acute manifestations of COVID-19, and the treatment received.” The recall bias might occur in this study and should be addressed.In methodology (page 5), “Pre-testing of the questionnaire was done, and supervised calls were made before the beginning of actual data collection.” Is there any validation of the questionnaire prior to the initiation of the study?The sample size for the study was calculated separately for mild to moderate cases and severe cases. Based on previous estimates, a 20% prevalence was taken for mild to moderate cases and the required sample size was 400. (14) For severe cases we assumed a 50% prevalence of Long COVID, and the sample size was calculated to be 100. A final sample of 487 individuals was analyzed (Figure 1) in the results (page 7). The sample size estimation was confusing to the readers and the occurrence of Long COVID for different severity was not specified. The data should be described in sufficient detail in Tables 1-3. The authors should present the possible associated factors for the occurrence of Long COVID prior to analysis using a logistic regression model.

Reviewers' comments:

Reviewer's Responses to Questions

**Comments to the Author**

1. Is the manuscript technically sound, and do the data support the conclusions?

Reviewer #1: Partly

2. Has the statistical analysis been performed appropriately and rigorously? 

Reviewer #1: I Don't Know

3. Have the authors made all data underlying the findings in their manuscript fully available?

Reviewer #1: No

4. Is the manuscript presented in an intelligible fashion and written in standard English?

Reviewer #1: Yes

5. Review Comments to the Author

Reviewer #1: In this cross-sectional single-site study, Singh et al describe the prevalence of persistent symptoms 4 weeks after COVID-19 diagnosis as determined by patient report on a telephone survey. While as the authors note this report adds to the minimal reporting of Long COVID in India to date, the findings are not novel, though they agree with prior reports. The study is limited by response bias and short follow-up time. Further, the authors do not specify the time period of data collection and when the patients were infected with SARS-CoV-2; given the different clinical outcomes of different variants this is important to know.

Specific comments:

Abstract:

Background section of abstract has some awkward language. I suggest defining Long COVID as “long-term symptoms after COVID-19”. I would specify that by “severe cases” you mean severe acute COVID-19, not severe long COVID.

Results section of abstract: I would change “significant number” to “higher number” as this is confusing with statistical significance.

Introduction:

Update numbers – 263 cases and 5 million deaths are no longer accurate numbers.

Methodology:

You specify that cases were diagnosed between April to September – but of what year? Can you describe the surges occurring at this time and the most common variants?

You mention that participants with missing phone numbers were scrubbed from the dataset. This is a source of bias and should be addressed. What was different about patients without phone numbers? Or those who refused to participate? Response bias should be noted.

Why was follow up done only at 4 weeks and not further out? Are there plans for more long term follow up? 4 weeks is quite short and many of these patients may recover in the following weeks.

Why was sample size included, if the goal was to simply describe the prevalence?

I’m confused about the use of multivariate logistic regression. The methods say that only the variables with p value <0.2 were included but Table 4 looks like all variables were included? Also, was occupation run as categorical or ordinal variable? Seems hard to assign an order to these categories, and being unemployed versus a student seems like very different backgrounds so it is unclear why these were lumped together. Finally, why was number of COVID-19 symptoms included as categorical variable instead of continuous? It seems that there is a huge difference in severity between having 1 versus 4 symptoms but these are lumped into same group.

Results:

Why were pregnant women excluded? There is no need given this is an observational study.

It is interesting that the previously twice vaccinated patients had more Long COVID at 4 weeks. You report prior infections and vaccination, but how many of these participants had both prior infection and vaccination? How long ago were the vaccinations? Of those who had prior COVID-19, did they have long covid after that infection too? What was different about participants who were vaccinated? Were participants with more comorbidities or older age more likely to be vaccinated, and that’s why they also were more likely to have long COVID?

The sentence reading “Females were 199…” should be re-written. “One hundred ninety nine (40.9%) participants were female, and the majority were… [specify graduates of what? College graduates? High school graduates?]”

6. PLOS authors have the option to publish the peer review history of their article (what does this mean?). If published, this will include your full peer review and any attached files.

Reviewer #1: No

---

## [Author Response · Author response to Decision Letter 0]

6 Nov 2022

Old title: Prevalence, characteristics, and predictors of Long COVID among diagnosed cases of COVID-19

New title: Characteristics and predictors of Long COVID among diagnosed cases of COVID-19

Additional Editor Comments and Response :

Comment: This study aims to estimate the prevalence and identify the characteristics and predictors of Long COVID among patients with acute COVID-19. Long COVID occur following acute COVID-19 and this condition develop during a particular time period, i.e., a new case, therefore, this study should determine the incidence rather than the prevalence. The median follow-up time of participants was 44 days in the results (page 7), however the follow-up time was too short for determining the occurrence of Long COVID.

Reply: 1 This study aims to estimate the prevalence and identify the characteristics and predictors of Long COVID among patients with acute COVID-19. Long COVID occur following acute COVID-19 and this condition develop during a particular time period, i.e., a new case, therefore, this study should determine the incidence rather than the prevalence. The median follow-up time of participants was 44 days in the results (page 7), however the follow-up time was too short for determining the occurrence of Long COVID. Thank you for the comments. We have removed the word “prevalence” from the manuscript and has edited the title also. 

The follow-up period was decided based on the definition of Long COVID given by National Institute for Health and Care Excellence (NICE) UK. The same definition is used by Govt. of India. According to NICE UK definition, Long COVID is classified as signs and symptoms that continue or develop after acute COVID‑19, including both ongoing symptomatic COVID‑19 (from 4 to 12 weeks) and post‑COVID‑19 syndrome (12 weeks or more).

We have followed up this cohort at 6 months and this data was not available at the time of submission to journal. In response to Editor and reviewer comments, we have added the 6 months follow-up data to this manuscript. (“Manuscript” file without track changes: (Results at Page 12, Lines:213-222 and Table 3) Thank you.

Comment 2: 2 In methodology (page 5), “these individuals were contacted through telephone after four weeks from the date of their COVID-19 diagnosis. After taking verbal consent, a detailed telephonic interview was conducted to record the socio-demographic details, past medical history including chronic disease and substance use, acute manifestations of COVID-19, and the treatment received.” The recall bias might occur in this study and should be addressed. Thank you for pointing this out. We have now discussed this possibility of Recall Bias in the limitation paragraph of the revised manuscript (“Manuscript” file without track changes: Page 19, Lines: 328-330) 

Since the data on COVID-19 diagnosis was taken from the hospital records, we do not expect any bias in estimating our primary objective. We also believe that since COVID-19 diagnosis was a potential life changing diagnosis, the patients might recall much of information, although the possibility of recall bias still exist. 

Reply: 2 In methodology (page 5), “these individuals were contacted through telephone after four weeks from the date of their COVID-19 diagnosis. After taking verbal consent, a detailed telephonic interview was conducted to record the socio-demographic details, past medical history including chronic disease and substance use, acute manifestations of COVID-19, and the treatment received.” The recall bias might occur in this study and should be addressed. Thank you for pointing this out. We have now discussed this possibility of Recall Bias in the limitation paragraph of the revised manuscript (“Manuscript” file without track changes: Page 19, Lines: 328-330) 

Since the data on COVID-19 diagnosis was taken from the hospital records, we do not expect any bias in estimating our primary objective. We also believe that since COVID-19 diagnosis was a potential life changing diagnosis, the patients might recall much of information, although the possibility of recall bias still exist. 

Comment 3: In methodology (page 5), “Pre-testing of the questionnaire was done, and supervised calls were made before the beginning of actual data collection.” Is there any validation of the questionnaire prior to the initiation of the study?

Reply 3: The questionnaire was adapted from the W.H.O Global COVID-19 Clinical Platform Case Report Form (CRF) for Post COVID condition (Post COVID-19 CRF). Thus, we did not do a separate validation. We have pre-tested the questionnaire and trained the data collectors for standardization and accuracy. 

Comment 4: 4 The sample size for the study was calculated separately for mild to moderate cases and severe cases. Based on previous estimates, a 20% prevalence was taken for mild to moderate cases and the required sample size was 400. (14) For severe cases we assumed a 50% prevalence of Long COVID, and the sample size was calculated to be 100. A final sample of 487 individuals was analyzed (Figure 1) in the results (page 7). The sample size estimation was confusing to the readers and the occurrence of Long COVID for different severity was not specified. The data should be described in sufficient detail in Tables 1-3. The authors should present the possible associated factors for the occurrence of Long COVID prior to analysis using a logistic regression model. Thank you for helping us improve the manuscript. We have rewritten the sample size paragraph with more clarity. (Page 6, Lines: 125-135) 

The primary objective was to estimate the proportion of COVID-19 patients who report the symptoms of Long COVID. We also planned to estimate this proportion separately for different severity of acute COVID-19. Hence, we did separate sample size estimation.

The results are now enriched with the addition of 6 months data. Figure 2 has been edited to highlight the finding of our primary objective.

The variables added to the logistic regression model was taken from the W.H.O Global COVID-19 Clinical Platform Case Report Form (CRF) for Post COVID condition (Post COVID-19 CRF). This was the reason we did not mention the variables separately in the original manuscript. We have now mentioned the choice of variables in Methodology section of the revised manuscript. (Page 6, Lines:143-146)

Reply 4: Thank you for helping us improve the manuscript. We have rewritten the sample size paragraph with more clarity. (Page 6, Lines: 125-135) 

The primary objective was to estimate the proportion of COVID-19 patients who report the symptoms of Long COVID. We also planned to estimate this proportion separately for different severity of acute COVID-19. Hence, we did separate sample size estimation.

The results are now enriched with the addition of 6 months data. Figure 2 has been edited to highlight the finding of our primary objective.

The variables added to the logistic regression model was taken from the W.H.O Global COVID-19 Clinical Platform Case Report Form (CRF) for Post COVID condition (Post COVID-19 CRF). This was the reason we did not mention the variables separately in the original manuscript. We have now mentioned the choice of variables in Methodology section of the revised manuscript. (Page 6, Lines:143-146)

Reviewer #1 Comments and Response

Comment 1: Reviewer #1: In this cross-sectional single-site study, Singh et al describe the prevalence of persistent symptoms 4 weeks after COVID-19 diagnosis as determined by patient report on a telephone survey. While as the authors note this report adds to the minimal reporting of Long COVID in India to date, the findings are not novel, though they agree with prior reports. The study is limited by response bias and short follow-up time. Further, the authors do not specify the time period of data collection and when the patients were infected with SARS-CoV-2; given the different clinical outcomes of different variants this is important to know.

Reply: 1 Reviewer #1: In this cross-sectional single-site study, Singh et al describe the prevalence of persistent symptoms 4 weeks after COVID-19 diagnosis as determined by patient report on a telephone survey. While as the authors note this report adds to the minimal reporting of Long COVID in India to date, the findings are not novel, though they agree with prior reports. The study is limited by response bias and short follow-up time. Further, the authors do not specify the time period of data collection and when the patients were infected with SARS-CoV-2; given the different clinical outcomes of different variants this is important to know. Thank you for the comments. Long COVID is now a research priority all over the world especially with the release of National Research Action Plan on Long COVID by the US Department of Health

and Human Services. The plan recognizes the need to have more studies on Long COVID and its risk factors from different geographical regions of the world. During the submission of this manuscript, there was hardly any well conducted study from India which used standard definitions and questionnaire. We believe the novelty in our research is that we bridge this evidence gap. 

The response bias is now discussed in the limitation section of the revised manuscript. (“Manuscript” file without track changes Page 19, Lines:332-334). Thank you for the insights.

We now have the 6 months follow up data of this cohort which was not available during the original submission to journal. We have added the 6-month data and revised the manuscript. 

The missing time period (year) was a typographical error, and this is corrected in the revised manuscript. Thank you for pointing this out. We have also added the reference to data on genetic variants of COVID-19 predominant in the community at the time conducting our study (Page 5, Lines:97-100, Reference No 20).

Comment 2: 2 Abstract: 

Background section of abstract has some awkward language. I suggest defining Long COVID as “long-term symptoms after COVID-19”. I would specify that by “severe cases” you mean severe acute COVID-19, not severe long COVID. Results section of abstract: I would change “significant number” to “higher number” as this is confusing with statistical significance. Thank you for the valuable inputs to improve our manuscript. We have incorporated all the suggestions in the revised manuscript.

Reply 2: Thank you for the valuable inputs to improve our manuscript. We have incorporated all the suggestions in the revised manuscript.

Comment 3: Introduction:

Update numbers – 263 cases and 5 million deaths are no longer accurate numbers.

Reply 3: The numbers are updated. Thank you.

Comment 4: Methodology:

You specify that cases were diagnosed between April to September – but of what year? Can you describe the surges occurring at this time and the most common variants?

Reply 4:Thank you for pointing out this mistake. We have added the year 2021. The surge occurring at this time was due to Delta variant (B.1.617.2) and we have described and added reference for the same in the methodology. (Page 5, Lines:97-100, Reference No 20).

Comment 5: Methodology:

You mention that participants with missing phone numbers were scrubbed from the dataset. This is a source of bias and should be addressed. What was different about patients without phone numbers? Or those who refused to participate? Response bias should be noted.

Reply 5: We agree with reviewer that there is a possibility of response bias. Since the COVID-19 hospital database we accessed did not have baseline characteristics of the patients and all data in the dataset came after the telephonic interview, we don’t have any meaningful data to make the comparison. We have discussed the same in limitations. (Page 19, Lines:332-334)

Comment 6: 6 Methodology:

Why was follow up done only at 4 weeks and not further out? Are there plans for more long term follow up? 4 weeks is quite short and many of these patients may recover in the following weeks. The 4 weeks follow-up was chosen based on the Long COVID definition given by National Institute for Health and Care Excellence (NICE) UK. The same definition is used by Govt. of India. (Reference 19)

During submission we did not had data on further follow-up. But now we are ready with the 6 months follow-up data and the same is added to the manuscript. (Page 12, Lines:213-222, Table 3) The choice of 6 months follow-up is also based on Long COVID definition by NICE.

Reply 6: The 4 weeks follow-up was chosen based on the Long COVID definition given by National Institute for Health and Care Excellence (NICE) UK. The same definition is used by Govt. of India. (Reference 19)

During submission we did not had data on further follow-up. But now we are ready with the 6 months follow-up data and the same is added to the manuscript. (Page 12, Lines:213-222, Table 3) The choice of 6 months follow-up is also based on Long COVID definition by NICE.

Comment 7: Methodology:

Why was sample size included if the goal was to simply describe the prevalence?

Reply 7: 7 Methodology:

Why was sample size included if the goal was to simply describe the prevalence? Thank you for the comment. We needed a rough estimate on how many participants to be followed up to get a meaningful estimate of incidence. The process of sample size calculation is rewritten based on comments from the Editor of the journal. (Page 6, Lines: 125-135)

Comment 8: Methodology:

I’m confused about the use of multivariate logistic regression. The methods say that only the variables with p value <0.2 were included but Table 4 looks like all variables were included? Also, was occupation run as categorical or ordinal variable? Seems hard to assign an order to these categories and being unemployed versus a student seems like very different backgrounds so it is unclear why these were lumped together. Finally, why was number of COVID-19 symptoms included as categorical variable instead of continuous? It seems that there is a huge difference in severity between having 1 versus 4 symptoms but these are lumped into same group.

Reply 8: 8 Methodology:

I’m confused about the use of multivariate logistic regression. The methods say that only the variables with p value <0.2 were included but Table 4 looks like all variables were included? Also, was occupation run as categorical or ordinal variable? Seems hard to assign an order to these categories and being unemployed versus a student seems like very different backgrounds so it is unclear why these were lumped together. Finally, why was number of COVID-19 symptoms included as categorical variable instead of continuous? It seems that there is a huge difference in severity between having 1 versus 4 symptoms but these are lumped into same group. Thank you for the comments. The plan was to use both statistics and clinical significance for including the variables in the multivariable logistic regression. Since most of the variables were clinically significant, they were retained except in case of collinearity. The line on p value < 0.2 is dropped to avoid confusion.

Occupation was run as categorical variable. They were lumped together to avoid excess subcategories and resultant decrease in statistical power.

The categorization of COVID-19 symptoms was based on previous literature. The grouping of the symptoms ensured that enough sample size is available in each category and adequate statistical power is available for analysis.

Comment 9: Results:

Why were pregnant women excluded? There is no need given this is an observational study.

Reply 9: Thank you for the comment. We agree that Pregnant women need not be excluded in this observational study. Since common Long COVID symptoms like fatigue is common in pregnancy, we took a decision on excluding pregnant women. Only 4 participants were excluded due to this reason.

Comment 10: Results:

It is interesting that the previously twice vaccinated patients had more Long COVID at 4 weeks. You report prior infections and vaccination, but how many of these participants had both prior infection and vaccination? How long ago were the vaccinations? Of those who had prior COVID-19, did they have long covid after that infection too? What was different about participants who were vaccinated? Were participants with more comorbidities or older age more likely to be vaccinated, and that’s why they also were more likely to have long COVID?

Reply 10: Only 16 participants had prior infection and at least one dose of vaccination. 

The date of last vaccination was collected but many of the participants could not report the exact date and due to high number of missing data the variable was dropped from analysis. But since COVID-19 positive patients cannot receive vaccination for 3 months post-infections, all the participants received vaccination prior to infection with COVID-19. 

Only 18 participants had history of past history of COVID-19. We did not have a variable asking these participants for Long COVID after the past infection. Since the numbers were small, we did not explore this further. 

It is likely that vaccinated individuals are different from unvaccinated with regards to age and comorbidities, but these variables were added and adjusted in the multivariable logistic regression to remove the confounding effects.

We thank you for the comments. Since the finding of vaccination increasing the odds of Long COVID was unusual, we have discussed this in detail by adding latest articles on this question, as well as discussed other possibilities of bias (Collider bias). We also checked for interaction of acute COVID-19 severity and COVID-19 vaccination on causing Long COVID and found no interaction. (Page 14, Lines:238-244) The same is reported with Odds ratio in the result section. In the Discussion section a full paragraph is dedicated to the discussion on these points and literature. (Page 18, Lines:304-319)

Comment 11: Results:

The sentence reading “Females were 199…” should be re-written. “One hundred ninety nine (40.9%) participants were female, and the majority were… [specify graduates of what? College graduates? High school graduates?]”

Reply 11: Thank you for the comments. We have edited these in the revised manuscript. (Page 8, Lines: 172-173)

---

## [Editor Report · Decision Letter 1]

24 Nov 2022

Characteristics and predictors of Long COVID among diagnosed cases of COVID-19

PONE-D-22-01446R1

Dear Dr. Singh,

We’re pleased to inform you that your manuscript has been judged scientifically suitable for publication and will be formally accepted for publication once it meets all outstanding technical requirements.

Kind regards,

Vipa Thanachartwet, M.D.

Academic Editor

PLOS ONE

Additional Editor Comments (optional):

We appreciate your efforts for the study and the authors have made a careful revision to the manuscript. All issues were revised according to the comments and suggestions.
---

## [Editor Report · Acceptance letter]

12 Dec 2022

PONE-D-22-01446R1 

 Characteristics and predictors of Long COVID among diagnosed cases of COVID-19 

Dear Dr. Singh:

I'm pleased to inform you that your manuscript has been deemed suitable for publication in PLOS ONE. Congratulations! Your manuscript is now with our production department. 

Kind regards, 

on behalf of

Associate Professor Vipa Thanachartwet 

Academic Editor

PLOS ONE